# Learning on One Mode: Addressing Multi-modality in Offline Reinforcement Learning

**Mianchu Wang**[*]     **Yue Jin**[*]     **Giovanni Montana**[*†]
[*]University of Warwick      [†]The Alan Turing Institute
{mianchu.wang, yue.jin.3, g.montana}@warwick.ac.uk

## Abstract

Offline reinforcement learning (RL) seeks to learn optimal policies from static datasets without interacting with the environment. A common challenge is handling multi-modal action distributions, where multiple behaviours are represented in the data. Existing methods often assume unimodal behaviour policies, leading to suboptimal performance when this assumption is violated. We propose weighted imitation Learning on One Mode (LOM), a novel approach that focuses on learning from a single, promising mode of the behaviour policy. By using a Gaussian mixture model to identify modes and selecting the best mode based on expected returns, LOM avoids the pitfalls of averaging over conflicting actions. Theoretically, we show that LOM improves performance while maintaining simplicity in policy learning. Empirically, LOM outperforms existing methods on standard D4RL benchmarks and demonstrates its effectiveness in complex, multi-modal scenarios.

## 1 Introduction

Offline reinforcement learning (RL) enables policy learning from static datasets, without active environment interaction, making it ideal for high-stakes applications like autonomous driving and robot manipulation (Levine et al., 2020; Ma et al., 2022; Wang et al., 2024a). A key challenge in offline RL is managing the discrepancy between the learned policy and the behaviour policy that generated the dataset. Small discrepancies can hinder policy improvement, while large discrepancies push the learned policy into uncharted areas, causing significant extrapolation errors and poor generalisation (Fujimoto et al., 2019; Yang et al., 2023). Addressing these challenges, existing research has proposed various solutions. Conservative approaches penalise actions that stray into out-of-distribution (OOD) regions (Yu et al., 2020; Kumar et al., 2020), while others regularise the policy by minimising its divergence from the behaviour policy, ensuring better fidelity to the dataset (Fujimoto & Gu, 2021; Wu et al., 2019). Another solution is weighted imitation learning, which aims to replicate actions from the dataset through supervised learning techniques (Mao et al., 2023; Peng et al., 2019).

Many real-world datasets introduce an additional challenge: multi-modal action distributions (Wang et al., 2024b; Chen et al., 2022; Zhou et al., 2020). These datasets are common in practice because they often integrate data from diverse sources, such as multiple policies, human demonstrations, or distinct exploration strategies. This diversity arises naturally in domains where the same task can be approached in various ways, leading to states with multiple valid but potentially conflicting actions. For instance, in autonomous driving, different driving styles — conservative versus aggressive — may lead to different but equally valid ways of navigating a road. Similarly, in robotic manipulation, an object can be grasped in various ways depending on the robot's approach, the object's position, and environmental constraints. This multi-modality is not an exception but rather a frequent occurrence in complex, real-world decision-making tasks, as systems often integrate experience from various sources to handle different scenarios. Thus, it becomes crucial to model and manage these multi-modal action distributions effectively.

Most offline RL approaches implicitly assume a unimodal action distribution, which can force policies to converge toward an average action that may not exist in the dataset, leading to degraded performance. This limitation is particularly evident in scenarios where policies fail to capture complex multi-modal distributions, instead collapsing into suboptimal or invalid averages, as studied by Cai et al. (2023); Wang et al. (2024b); Yang et al. (2022b) and illustrated in Appendix A. Recent work has attempted to address this by modelling the full multi-modal action distribution using expressive

generative models such as GANs, VAEs, and diffusion models (Zhou et al., 2020; Chen et al., 2022; Wang et al., 2023). However, these models often overcomplicate the learning process by capturing the entire action distribution, which is unnecessary when only a subset of the modes is relevant for optimal decision-making.

We propose a simpler yet effective approach: *Weighted Imitation Learning on One Mode (LOM)*. Our insight is that learning from a single mode — the one with the highest expected return — is sufficient to generate optimal actions. Instead of modelling the full multi-modal distribution, LOM identifies and focuses on the most promising mode for each state.

LOM operates through three key steps (illustrated in Figure 1). First, it models the behaviour policy as a Gaussian mixture model (GMM), capturing the inherent multi-modality in the action space. Each mode in the GMM represents a distinct cluster of actions associated with a state. Second, a novel hyper Q-function is introduced to evaluate the expected return of each mode, enabling the dynamic selection of the most advantageous one. Finally, LOM performs weighted imitation learning on the actions from the selected mode, ensuring that the learned policy focuses on the most beneficial actions while retaining the simplicity of unimodal policies. This targeted, mode-specific learning strategy simplifies the policy learning process while maintaining or even enhancing performance in multi-modal scenarios. By dynamically selecting the optimal mode for each state, LOM achieves robust results with reduced complexity.

This paper has four key contributions: (1) we propose LOM, a novel weighted imitation learning method designed to address the multi-modality problem in offline RL; (2) we introduce hyper Q-functions and hyper-policies for evaluating and selecting action modes; (3) we provide theoretical guarantees of consistent performance improvements over both the behaviour policy and the optimal action mode; and (4) we empirically demonstrate that LOM outperforms state-of-the-art (SOTA) offline RL methods across various benchmarks, particularly in multi-modal datasets.

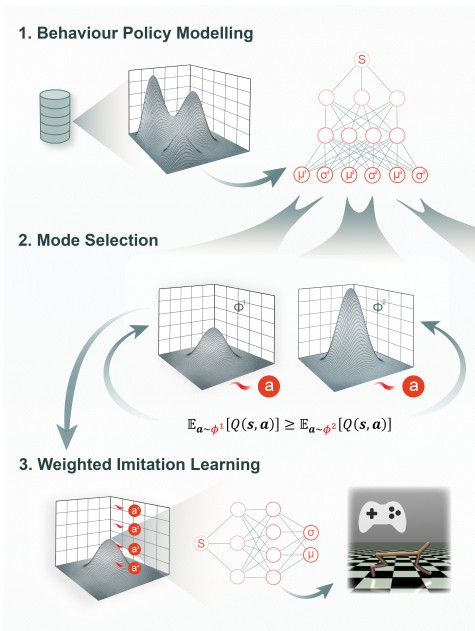

Figure 1: The three steps of LOM. (1) Learn a network producing the parameters of a GMM to model the behaviour policy. (2) Evaluate each mode via the expected return of its actions and then select the optimal mode $\phi^1$. (3) Sample actions from $\phi^1$ for weighted imitation learning.

## 2 RELATED WORK

**Offline RL** The primary challenge in offline RL lies in managing the distribution shift between the behaviour policy, which generated the offline dataset, and the learned policy. This shift can cause the learned policy to produce actions that are not well-represented in the offline dataset, leading to inaccurate value function estimations and degraded performance (Levine et al., 2020; Fujimoto et al., 2019). To mitigate the risks associated with OOD actions, one approach involves using conservative value functions, which penalise actions that deviate from the behaviour policy (Fujimoto et al., 2019; Kumar et al., 2020; Bai et al., 2022; Sun et al., 2022; An et al., 2021). Another strategy focuses on regularising the learned policy to ensure its proximity to the behaviour policy. This regularisation can be measured using metrics such as mean squared error (MSE) (Beeson & Montana, 2024; Fujimoto & Gu, 2021) or more sophisticated metrics like the Wasserstein distance, Jensen-Shannon divergence (Yang et al., 2022b; Wu et al., 2019), and weighted MSE (Ma et al., 2024). These methods aim to keep the learned policy within a safer, well-understood operational space, reducing the likelihood of selecting OOD actions. For a comprehensive review of value penalties and policy regularisation techniques, we refer readers to (Wu et al., 2019). In contrast to these approaches, LOM tackles the distribution shift problem by focusing on a single, optimal mode of the behaviour policy, thereby reducing the risk of OOD actions while still allowing for significant policy improvement.

**Weighted Imitation Learning**    Our proposed method contributes to the growing field of weighted imitation learning, which revolves around two fundamental questions: which actions should be imitated and how should these actions be weighted (Brandfonbrener et al., 2021). Most existing methods imitate actions directly from the dataset (Peng et al., 2019; Wang et al., 2018; Nair et al., 2021; Wang et al., 2020), while some approaches (Siegel et al., 2020; Mao et al., 2023) suggest imitating actions from the most recent policies. These methods typically weigh actions based on their advantage, conditioned either on the behaviour policy (Peng et al., 2019; Wang et al., 2018) or on the most recently updated policy (Nair et al., 2021; Wang et al., 2020; Siegel et al., 2020; Mao et al., 2023). However, a key limitation of these approaches is their underlying assumption that the imitated actions are drawn from a unimodal Gaussian distribution, an oversimplification that often fails in complex, real-world datasets where multi-modal distributions are common (Lynch et al., 2020; Yang et al., 2022b). LOM addresses this limitation by forgoing the unimodal assumption, instead extracting a highly-rewarded Gaussian mode from the behaviour policy.

**Multi-modality in Offline RL**    Offline RL datasets are often collected from multiple, unknown policies, leading to states with several valid action labels, which may conflict with one another. Most existing methods employ a unimodal Gaussian policy, which inadequately captures the inherent multi-modality in such datasets. To address this limitation, PLAS (Zhou et al., 2020) decodes variables sampled from a VAE latent space, while LAPO (Chen et al., 2022) leverages advantage-weighted latent spaces for further policy optimisation. Additionally, Yang et al. (2022b) demonstrate that GANs can model multiple action modes, and GOPlan (Wang et al., 2024b) extends this by applying exponentiated advantage weighting to highlight highly-rewarded modes. DAWOG (Wang et al., 2024a) and DMPO (Osa et al., 2023) separate the state-action space and learn a conditioned Gaussian policy or a mixture of deterministic policies to solve the regions individually. Techniques such as normalising flows and diffusion models can also model multi-modal distributions by progressively transforming an initial distribution into the target distribution (Wang et al., 2023; Akimov et al., 2022). However, these techniques are computationally intensive, making their inference processes inefficient. Beyond direct policy improvements, approaches like TD3+RKL (Cai et al., 2023) exploit the mode-seeking property of reverse KL-divergence to narrow action coverage, while BAW (Yang et al., 2022a) filters actions with values in the leftmost quantile. Despite these advances, most methods attempt to capture or model the entire multi-modal distribution, leading to increased complexity. In contrast, LOM simplifies the learning process by extracting and focusing on a single, optimal mode from the behaviour policy, effectively addressing the multi-modality challenge without the need to model the entire distribution.

## 3    PRELIMINARIES

To lay the foundation for our method, we first introduce the Markov Decision Process (MDP) and the concept of weighted imitation learning. An MDP is defined by the tuple $\mathcal{M} = \langle \mathcal{S}, \mathcal{A}, \mathcal{P}, r, \gamma \rangle$, where $\mathcal{S}$ is the state space, $\mathcal{A}$ the action space, and $\mathcal{P}$ the transition dynamics. The function $r$ represents the reward, and $\gamma$ is the discount factor. The objective in RL is to learn a policy $\pi$ that maximises the expected discounted return:

$$J(\pi) = \mathbb{E}_{\tau \sim p_\pi} \left[ \sum_{t=0}^{T} \gamma^t r(s_t, a_t) \right],$$

where $\tau = \{s_0, a_0, \ldots, s_T, a_T\}$ is a trajectory sampled under policy $\pi$. The state-action value function $Q^\pi(s_t, a_t)$ represents the expected return starting from state $s_t$, taking action $a_t$, and following $\pi$ thereafter:

$$Q^\pi(s_t, a_t) = \mathbb{E}_{s_{t+1}, a_{t+1}, \cdots \sim \pi} \left[ \sum_{l=t}^{T} \gamma^l r(s_l, a_l) \right]. \tag{1}$$

The value function is defined as $V^\pi(s_t) = \mathbb{E}_{a_t \sim \pi(\cdot|s_t)}[Q^\pi(s_t, a_t)]$, and the advantage function is $A^\pi(s_t, a_t) = Q^\pi(s_t, a_t) - V^\pi(s_t)$.

In offline RL, the agent learns exclusively from a fixed dataset $\mathcal{D}$ collected by one or more behaviour policies, denoted as $\pi_b$, without further interaction with the environment (Levine et al., 2020). Our approach is based on weighted imitation learning, where the objective is to optimize

$$J(\pi) = \mathbb{E}_{s \sim d_{\pi_b}, a \sim \pi_b(\cdot|s)} \left[ \exp \left( \frac{1}{\beta} A^{\pi_b}(s, a) \right) \log \pi(a \mid s) \right],$$

where $\beta$ is a hyper-parameter. This formulation encourages learning a policy $\pi$ that imitates actions from $\mathcal{D}$, weighted by the exponentiated advantage of the behaviour policy $\pi_b$, thereby implicitly constraining the KL-divergence between $\pi$ and $\pi_b$ (Wang et al., 2018). Recent works have extended this approach by imitating actions from the current learned policy (Siegel et al., 2020; Mao et al., 2023), or weighting actions based on the advantage conditioned on the current policy (Nair et al., 2021; Wang et al., 2020).

## 4 METHOD

In this section, we present the LOM method, specifically designed for offline RL with heterogeneous datasets. We begin by modelling the behaviour policy using a Gaussian Mixture Model (GMM) to capture the inherent multi-modality. Following this, we formalise the problem using a hyper-Markov decision process (H-MDP), where a hyper-policy dynamically selects the most promising mode for each state based on expected returns. Next, we propose a hyper Q-function to evaluate the hyper-policy and introduce a greedy hyper-policy for mode selection at each step for weighted imitation learning. Finally, we prove that the resulting policy of our LOM method outperforms both the behaviour policy and a greedy policy.

### 4.1 MULTI-MODAL BEHAVIOURAL POLICY

To address the challenge of multi-modal action distributions in offline RL, we model the behaviour policy as a Gaussian Mixture Model (GMM). GMMs are flexible in representing complex distributions and can approximate any density function with sufficient components (McLachlan & Basford, 1988), making them well-suited for capturing the heterogeneous nature of offline datasets that often arise from multiple policies or exploration strategies. The behaviour policy $\pi_b(a \mid s)$ is expressed as:

$$\pi_b(a \mid s) = \sum_{i=1}^{M} \alpha^i(s)\phi^i(a \mid s), \tag{2}$$

where $\alpha^i(s)$ represents the mixing coefficients for each mode $i$, and $\phi^i(a \mid s)$ denotes the Gaussian component for mode $i$ at state $s$. Each mode captures a distinct cluster of actions associated with the state. By using a GMM, our model captures the diverse modes present in real-world offline RL datasets, where multiple valid strategies may coexist for the same task. This approach preserves the richness of the original data while avoiding the limitations of unimodal policy approximations.

### 4.2 HYPER-MARKOV DECISION PROCESS

To extend the traditional MDP framework introduced in Section 3 and formulate the offline RL problem with a multi-modal behaviour policy, we introduce the *Hyper-Markov Decision Process* (H-MDP). The H-MDP accounts for multiple policy modes, extending the standard MDP to a higher level of abstraction. Formally, the H-MDP is defined as $\mathcal{M}_H = \langle \mathcal{S}, \mathcal{A}, \mathcal{P}, r, \gamma, \Omega, \mathcal{A}_H, \mathcal{P}_H, r_H \rangle$, where $\mathcal{S}$, $\mathcal{A}$, $\mathcal{P}$, $r$, and $\gamma$ retain their standard meanings from the MDP. The novel components introduced are: $\Omega = \{\phi^1, \ldots, \phi^M\}$, representing the Gaussian modes, and the *hyper-action space* $\mathcal{A}_H = \{1, \ldots, M\}$, which indexes these modes.

The transition dynamics $\mathcal{P}_H$ conditioned on the hyper-action $u_t \in \mathcal{A}_H$ and the reward function $r_H$ for $u_t$ are defined as:

$$\mathcal{P}_H(s_{t+1} \mid s_t, u_t) = \mathbb{E}_{a_t \sim \phi^{u_t}(a \mid s_t)} \left[ \mathcal{P}(s_{t+1} \mid s_t, a_t) \right],$$

$$r_H(s_t, u_t) = \mathbb{E}_{a_t \sim \phi^{u_t}(a \mid s_t)} \left[ r(s_t, a_t) \right].$$

At each time step, the agent selects a Gaussian mode $u$ based on a *hyper-policy* $\zeta$, which maps states to a probability distribution over modes, $\zeta(u \mid s) : \mathcal{S} \times \mathcal{A}_H \to [0, 1]$. The hyper-policy determines the likelihood of selecting each mode in a given state. The agent then selects an action $a$ according to the Gaussian distribution $\phi^u(a \mid s)$. Consequently, the action follows a *composite policy* $\pi_\zeta$, which corresponds to the hyper-policy $\zeta$ and can be expressed as:

$$\pi_\zeta(a \mid s) = \sum_{u \in \mathcal{A}_H} \zeta(u \mid s)\phi^u(a \mid s), \tag{3}$$

In the H-MDP, the agent's objective is to learn a hyper-policy $\zeta$ that maximises the expected discounted return:

$$J(\zeta) = \mathbb{E}_{\tau \sim p_\zeta} \left[ \sum_{t=0}^{T} \gamma^t r_H(s_t, u_t) \right], \tag{4}$$

where $p_\zeta$ is the distribution of state-mode trajectories $\tau = (s_t, u_t, s_{t+1}, u_{t+1}, \dots)$ induced by following the hyper-policy $\zeta$.

### 4.3 HYPER Q-FUNCTION

To evaluate the quality of selecting a mode $u$ in a given state $s$, we define the *hyper Q-function* $Q_H^\zeta(s, u)$. This function quantifies the expected return when choosing mode $u$ at state $s$ and subsequently following the hyper-policy $\zeta$. Formally, the hyper Q-function is defined as:

$$Q_H^\zeta(s, u) = \mathbb{E}_{s_{t+1}, u_{t+1}, \dots \sim \zeta} \left[ \sum_{t=0}^{T} \gamma^t r_H(s_t, u_t) \mid s_0 = s, u_0 = u \right]. \tag{5}$$

This expectation is taken over future states and modes encountered while following the hyper-policy $\zeta$ after selecting mode $u$ in state $s$. The term $r_H(s_t, u_t)$ denotes the expected reward for selecting mode $u_t$ in state $s_t$, with $\gamma$ being the discount factor.

**Proposition 1.** *The hyper Q-function can be linked to the standard value function $Q^\pi(s, a)$ via:*

$$Q_H^\zeta(s, u) = \mathbb{E}_{a \sim \phi^u(\cdot \mid s)} \left[ Q^{\pi_\zeta}(s, a) \right]. \tag{6}$$

*The proof can be found in Appendix B.1.*

This proposition establishes a critical relationship between the hyper Q-function and the standard Q-function. It shows that the expected return for selecting mode $u$ (i.e., the hyper-action) is equivalent to the expected value of actions sampled from the Gaussian component $\phi^u(a \mid s)$, evaluated under the composite policy $\pi_\zeta$. This result is particularly important when dealing with multi-modal behaviour policies, as it allows the agent to effectively compare and evaluate different modes $u$ based on their associated action distributions $\phi^u(a \mid s)$.

### 4.4 MODE SELECTION

Since the hyper Q-function evaluates the expected return after selecting mode $u$ at a state, we now propose a greedy hyper-policy that improves upon the behaviour policy by selecting the most advantageous mode at each state based on the hyper Q-function.

In correspondence with the multi-modal behaviour policy $\pi_b$ defined in Eq. 2, we introduce the *behavioural hyper-policy*, denoted as $\zeta_b$, which reflects the mode-selection strategy implicit in the behaviour policy that generated the offline dataset. Specifically, $\zeta_b(u \mid s) = \alpha^u(s)$, where $\alpha^u(s)$ is the mixing coefficient for the $u$-th Gaussian component in the GMM at state $s$. This formulation establishes the connection between the original behaviour policy $\pi_b$ and the H-MDP framework, allowing us to represent the behaviour policy in terms of mode selection.

To improve upon the behavioural hyper-policy $\zeta_b$, we define a *greedy hyper-policy* $\zeta_g$, which selects the mode that maximises the hyper Q-function $Q_H^{\zeta_b}(s, u)$ for each state:

$$\zeta_g(s) = \arg \max_{u \in \mathcal{A}_H} Q_H^{\zeta_b}(s, u). \tag{7}$$

This greedy hyper-policy ensures that the agent selects the mode with the highest expected return at each state, thereby improving the overall policy performance relative to the behaviour policy.

**Theorem 1.** *The composite policy induced by the greedy hyper-policy improves upon the behaviour policy:*

$$V^{\pi_{\zeta_g}}(s) \geq V^{\pi_b}(s), \quad \forall s \in \mathcal{S}. \tag{8}$$

*The proof can be found in Appendix B.2.*

This improvement property guaranteed by Theorem 1 forms the foundation of our LOM algorithm. It ensures that by systematically selecting modes with the highest expected return at each state, the agent's composite policy will, over time, perform at least as well as the original behaviour policy.

### 4.5 WEIGHTED IMITATION LEARNING ON ONE MODE

Based on the greedy hyper-policy $\zeta_g$, we obtain the corresponding greedy policy $\pi_{\zeta_g}$, which, according to Theorem 1, is not worse than $\pi_b$. However, rather than simply imitating $\pi_{\zeta_g}$, we apply weighted imitation learning to further improve the policy. Specifically, we aim to find a policy $\pi$ that maximises the expected improvement $\eta(\pi) = J(\pi) - J(\pi_{\zeta_g})$. The expected improvement $\eta(\pi)$ can be expressed in terms of the advantage $A^{\pi_{\zeta_g}}(s, a)$ (Kakade & Langford, 2002; Schulman et al., 2015): $\eta(\pi) = \mathbb{E}_{s \sim d_\pi(\cdot), a \sim \pi(\cdot|s)}[A^{\pi_{\zeta_g}}(s, a)]$. However, in offline RL, state samples from the distribution $d_\pi(s)$ are unavailable. Instead, we approximate the expected improvement by using an estimate $\hat{\eta}$ (Kakade & Langford, 2002): $\hat{\eta}(\pi) = \mathbb{E}_{s \sim d_{\pi_{\zeta_g}}(\cdot), a \sim \pi(\cdot|s)}[A^{\pi_{\zeta_g}}(s, a)]$. This approximation provides a good estimate of $\eta(\pi)$ if $\pi$ and $\pi_{\zeta_g}$ are close in terms of KL-divergence (Schulman et al., 2015). As a result, we can formulate the following constrained policy optimisation problem:

$$\arg\max_\pi \sum_s d_{\pi_{\zeta_g}}(s) \sum_a \pi(a \mid s) A^{\pi_{\zeta_g}}(s, a),$$
$$\text{s.t.} \sum_s d_{\pi_{\zeta_g}}(s) D_{KL}(\pi(\cdot \mid s) || \pi_{\zeta_g}(\cdot \mid s)) \le \epsilon. \tag{9}$$

Solving the corresponding Lagrangian leads to the optimal policy $\pi^*$: $\pi^*(a \mid s) = \frac{1}{Z(s)} \pi_{\zeta_g}(a \mid s) \exp\left(\frac{1}{\beta} A^{\pi_{\zeta_g}}(s, a)\right)$. We then learn a policy by minimising the KL-divergence to this optimal policy, resulting in the weighted imitation learning objective:

$$\arg\min_\pi \mathbb{E}_{s \sim d_{\pi_b}(\cdot)}[D_{KL}(\pi^*(\cdot \mid s) || \pi(\cdot \mid s))] \tag{10}$$

$$= \arg\max_\pi \mathbb{E}_{s \sim d_{\pi_b}(\cdot), a \sim \pi_{\zeta_g}(\cdot|s)} \left[\exp\left(\frac{1}{\beta} A^{\pi_{\zeta_g}}(s, a)\right) \log \pi(a \mid s)\right]. \tag{11}$$

In this framework, the actions to be imitated are sampled from $\pi_{\zeta_g}$, a unimodal Gaussian policy, thereby addressing the multi-modality issue. Furthermore, actions within this mode are weighted according to their advantage, encouraging the policy to concentrate on the highly-rewarded actions. As shown in Theorem 2, the policy learned by LOM has a theoretical advantage over both $\pi_{\zeta_g}$ and the original behaviour policy $\pi_b$.

**Theorem 2.** *The LOM algorithm learns a policy $\pi_L$ that is at least as good as both the composite policy induced by the greedy hyper-policy $\pi_{\zeta_g}$ and the behaviour policy $\pi_b$. Specifically, for all states $s \in \mathcal{S}$:*

$$V^{\pi_L}(s) \ge V^{\pi_{\zeta_g}}(s) \ge V^{\pi_b}(s). \tag{12}$$

*The proof is provided in Appendix B.3.*

This theorem implies that our method introduces a novel two-step improvement process. LOM achieves further policy improvement compared to one-step algorithms while requiring less off-policy evaluation than multi-step algorithms (Brandfonbrener et al., 2021).

**Theorem 3.** *The LOM algorithm learns a policy $\pi_L$, whose improvement upon the composite policy induced by the greedy hyper-policy $\pi_{\zeta_g}$ is bounded. For all $s \in \mathcal{S}$, we have*

$$V^{\pi_L}(s) - V^{\pi_{\zeta_g}}(s) \ge \frac{1}{1-\gamma}\hat{\eta}(\pi_L) - \frac{A_{\max}}{1-\gamma}\sqrt{\frac{1}{2}D_{KL}(d_{\pi_L} || d_{\pi_{\zeta_g}})}, \tag{13}$$

*where $A_{\max} = \max_{s,a} |A^{\pi_{\zeta_g}}(s, a)|$. The proof is provided in Appendix B.4.*

The bound shows that the learned policy $\pi_L$ is guaranteed to perform at least as well as the baseline policy $\pi_{\zeta_g}$, with the performance improvement quantified by the maximal advantage and reduced by a penalty term proportional to the divergence in their state visitation distributions — emphasizing that maximising advantage while minimising divergence leads to better performance.

## 5 PRACTICAL ALGORITHM

In this section, we present the practical implementation of the LOM algorithm. First, we model the multi-modal behaviour policy using a mixture density network, as described in Section 5.1. In Section 5.2, we outline the mode selection process, which involves learning the hyper Q-function for the behavioural hyper-policy. This hyper Q-function is then used to derive the greedy hyper-policy $\zeta_g$ and the corresponding greedy policy $\pi_{\zeta_g}$, which selects a single mode. Finally, in Section 5.3, we describe how to learn a policy through weighted imitation learning on the selected mode.

---

**Algorithm 1** Weighted imitation learning on one mode (LOM).

---

**Initialise:** A MDN behaviour policy $\pi_\rho$ with parameter $\rho$, a target policy $\pi_\theta$ with parameter $\theta$, a value function $Q_\psi$ with parameter $\psi$ and its slowly-updated copy $Q_{\psi^-}$, a behaviour hyper Q-function $Q_\phi$ with parameter $\phi$; an offline dataset $\mathcal{D}$.
1: # Learn the MDN behaviour policy.
2: **for** $i = 1, ..., I_M$ **do**
3:    Update $\rho$ by minimising $\mathcal{L}(\rho) = \mathbb{E}_{(s_t,a_t)\sim\mathcal{D}}[\log \pi_\rho(a_t \mid s_t)]$.    ▷ Eq. 15.
4: **end for**
5: **for** $i = 1, \ldots, I_G$ **do**
6:    Sample transitions $\boldsymbol{\tau} = \{s_t, a_t, r_t, s_{t+1}, a_{t+1}\} \sim \mathcal{D}$.
7:    # Learn the value function.
8:    Update $\psi$ by minimising $\mathcal{L}(\psi) = \mathbb{E}_{\boldsymbol{\tau}}[(Q_\psi(s_t, a_t) - (r_t + \gamma Q_{\psi^-}(s_{t+1}, a_{t+1}))^2]$.    ▷ Eq. 16
9:    # Learn behavioural hyper Q-function.
10:   Update $\phi$ by minimising $\mathcal{L}(\phi) = \mathbb{E}_{s_t\sim\mathcal{D}, u\sim\text{Uniform}(\mathcal{A}_H)}[(Q_\phi(s_t, u) - \mathbb{E}_{a'_t\sim\phi^u(\cdot\mid s)}[Q_\psi(s_t, a'_t)])^2]$
11:   # Learn the target policy.
12:   Select the optimal mode $u = \arg\max_{u\in\{1,\ldots,M\}} Q_\phi(s_t, u)$
13:   Sample action from the optimal mode $\hat{a}_t \sim \phi^u(s_t)$
14:   Get the mean of the mode $\bar{a}_t = \phi^u_\mu(s_t)$    ▷ The mean of the mode
15:   Estimate the advantage $A(s_t, \hat{a}_t) = Q_\psi(s_t, \hat{a}_t) - Q_\psi(s_t, \bar{a}_t)$
16:   Update $\theta$ by minimising $\mathcal{L}(\theta) = -\mathbb{E}_{s_t,\hat{a}_t}[\exp(A(s_t, \hat{a}_t))\log \pi_\theta(\hat{a}_t \mid s_t)]$.    ▷ Eq. 18
17:   # Update the target network.
18:   **if** $i$ mode update_delay $= 0$ **then**
19:      $\psi^- \leftarrow \rho\psi^- + (1 - \rho)\psi$
20:   **end if**
21: **end for**

---

## 5.1 BEHAVIOUR POLICY MODELLING

At the start of the LOM algorithm, we model the behaviour policy using a mixture density network (MDN) (Bishop, 1994). The MDN receives a state $s$ and represents the resulting action distribution with a GMM, including its mixing coefficients, locations, and scales. We index the Gaussian components as the hyper-actions, ranging from 1 to $M$.

In the algorithm, we learn a neural network $\pi_\rho(a \mid s)$ parameterised by $\rho$ to estimate the behaviour policy. The network produces the parameters $\{z_i^\mu, z_i^\sigma, z^{\alpha^i}\}_{i=1}^M$, which are used to estimate the probability density function for each policy mode:

$$\phi^i(a \mid s) = \frac{1}{\sqrt{2\pi}\sigma_i(s)} \exp\left(-\frac{||a - \mu_i(s)||^2}{2\sigma_i(s)^2}\right), \tag{14}$$

where $\mu_i(s) = z_i^\mu$, and $\sigma_i(s) = \exp(z_i^\sigma)$. The mixing coefficient $\alpha^i(s)$ is processed through a softmax function: $\alpha^i(s) = \exp(z^{\alpha^i})/\sum_{j=1}^M \exp(z^{\alpha^j})$. The parameters of the network are updated by minimising the negative log-likelihood:

$$\mathcal{L}(\rho) = -\mathbb{E}_{(s,a)\sim\mathcal{D}}\left[\log \pi_\rho(a \mid s)\right]. \tag{15}$$

LOM trains the MDN until convergence, after which the parameter $\rho$ is fixed for the remainder of the algorithm.

## 5.2 HYPER Q-FUNCTION LEARNING

We learn the hyper Q-function of the behavioural hyper-policy based on Proposition 1. Specifically, we first learn a Q-function, $Q^{\pi_b}$, of the behaviour policy, and then estimate its expectation over actions sampled from mode $u$. We instantiate $Q^{\pi_b}$ with a neural network $Q_\psi$ parameterised by $\psi$ and updated to minimising the TD error:

$$\mathcal{L}(\psi) = \mathbb{E}_{(s_t,a_t,r_t,s_{t+1},a_{t+1})\sim\mathcal{D}}\left[(Q_\psi(s_t, a_t) - (r_t + \gamma Q_{\psi^-}(s_{t+1}, a_{t+1})))^2\right]. \tag{16}$$

where $Q_{\psi^-}$ is a slowly updated copy of $Q_\psi$. Following Proposition 1, we learn $Q_H^{\zeta_b}$, a hyper Q-function of the behavioural hyper-policy, which is denoted as $Q_\phi$ parameterised by $\phi$. The optimisation objective of learning $Q_\phi$ is:

$$\mathcal{L}(\phi) = \mathbb{E}_{s_t\sim\mathcal{D}, u\sim\text{Uniform}(\mathcal{A}_H)}\left[(Q_\phi(s_t, u) - \mathbb{E}_{a_t\sim\phi^u(\cdot\mid s)}[Q_\psi(s_t, a_t)])^2\right]. \tag{17}$$

With this hyper Q-function, we get a greedy hyper-policy $\zeta_g(s) = \arg\max_{u\in\mathcal{A}_H} Q_\phi(s, u)$. Corresponding to $\zeta_g(s)$, we get the greedy policy with one mode, $\pi_{\zeta_g}(a \mid s) = \phi^{\zeta_g(s)}(a \mid s)$.

Table 1: Averaged normalised scores on D4RL benchmarks over 4 random seeds. 0 represents the performance of a random policy and 100 represents the performance of an expert policy. Standard deviations are provided as subscripts for multi-modal handling algorithms. The highest mean is highlighted in bold.

| Dataset | Env | OneStep | AWAC | CQL | TD3BC | LAPO | DMPO | WCGAN | WCVAE | LOM |
|---|---|---|---|---|---|---|---|---|---|---|
| medium | halfcheetah | 50.4 | 47.9 | 47.0 | 48.3 | $45.9_{\pm 0.3}$ | $47.5_{\pm 0.4}$ | $48.2_{\pm 1.3}$ | $50.5_{\pm 1.1}$ | $\mathbf{51.0}_{\pm 0.7}$ |
| | hopper | 87.5 | 59.8 | 53.0 | 59.3 | $51.6_{\pm 3.2}$ | $71.2_{\pm 6.5}$ | $78.6_{\pm 2.4}$ | $89.0_{\pm 2.0}$ | $\mathbf{100.8}_{\pm 1.4}$ |
| | walker2d | 84.8 | 83.1 | 73.3 | 83.7 | $80.7_{\pm 0.8}$ | $79.4_{\pm 4.7}$ | $82.4_{\pm 1.7}$ | $85.0_{\pm 1.5}$ | $\mathbf{85.1}_{\pm 0.7}$ |
| medium-replay | halfcheetah | 42.7 | 44.8 | 45.5 | 44.6 | $44.7_{\pm 0.3}$ | $45.2_{\pm 0.8}$ | $42.3_{\pm 4.2}$ | $45.0_{\pm 0.9}$ | $\mathbf{48.8}_{\pm 0.7}$ |
| | hopper | 98.5 | 69.8 | 88.7 | 60.9 | $58.6_{\pm 3.8}$ | $89.2_{\pm 8.1}$ | $90.3_{\pm 3.1}$ | $99.0_{\pm 2.1}$ | $\mathbf{99.2}_{\pm 1.1}$ |
| | walker2d | 61.7 | 78.1 | 81.8 | 81.8 | $71.7_{\pm 5.2}$ | $82.1_{\pm 3.8}$ | $72.6_{\pm 4.0}$ | $74.0_{\pm 3.1}$ | $\mathbf{84.8}_{\pm 1.0}$ |
| medium-expert | halfcheetah | 75.1 | 64.9 | 75.6 | 90.7 | $\mathbf{93.0}_{\pm 1.0}$ | $91.1_{\pm 3.4}$ | $76.5_{\pm 3.1}$ | $80.0_{\pm 1.7}$ | $92.7_{\pm 1.3}$ |
| | hopper | 108.6 | 100.1 | 105.6 | 98.0 | $105.2_{\pm 4.7}$ | $78.4_{\pm 19.0}$ | $110.0_{\pm 2.4}$ | $109.0_{\pm 1.6}$ | $\mathbf{110.1}_{\pm 1.4}$ |
| | walker2d | 111.3 | 110.0 | 107.9 | 110.1 | $111.1_{\pm 0.2}$ | $109.9_{\pm 0.4}$ | $99.7_{\pm 1.0}$ | $111.0_{\pm 0.7}$ | $\mathbf{111.3}_{\pm 0.8}$ |
| expert | halfcheetah | 88.2 | 81.7 | 96.3 | 96.7 | $95.9_{\pm 0.2}$ | $\mathbf{97.0}_{\pm 1.0}$ | $90.7_{\pm 1.7}$ | $89.0_{\pm 0.8}$ | $95.2_{\pm 0.4}$ |
| | hopper | 106.9 | 109.5 | 96.5 | 107.8 | $106.7_{\pm 3.6}$ | $93.6_{\pm 15.1}$ | $107.3_{\pm 1.8}$ | $108.0_{\pm 1.3}$ | $\mathbf{111.0}_{\pm 0.8}$ |
| | walker2d | 110.7 | 110.1 | 108.5 | 110.2 | $\mathbf{112.2}_{\pm 0.1}$ | $111.4_{\pm 0.3}$ | $109.3_{\pm 1.4}$ | $111.0_{\pm 1.4}$ | $109.3_{\pm 1.1}$ |
| full-replay | halfcheetah | 64.7 | 66.6 | 73.6 | 71.7 | $74.2_{\pm 1.3}$ | $71.4_{\pm 1.2}$ | $64.2_{\pm 3.1}$ | $66.0_{\pm 3.4}$ | $\mathbf{76.6}_{\pm 1.2}$ |
| | hopper | 69.8 | 100.1 | 98.2 | 76.8 | $100.0_{\pm 3.3}$ | $101.5_{\pm 5.9}$ | $80.2_{\pm 5.3}$ | $86.5_{\pm 4.4}$ | $\mathbf{102.0}_{\pm 2.7}$ |
| | walker2d | 67.1 | 78.3 | 92.7 | 90.2 | $96.2_{\pm 2.8}$ | $95.4_{\pm 1.8}$ | $53.0_{\pm 5.4}$ | $72.0_{\pm 3.7}$ | $\mathbf{97.9}_{\pm 0.9}$ |

## 5.3 Policy Learning

Finally, we learn the policy $\pi_\theta$ by maximising the objective:

$$J(\theta) = \mathbb{E}_{s \sim \mathcal{D}, a \sim \pi_{\zeta_g}(\cdot|s)} \left[ \exp_{clip}(\frac{1}{\beta} A^{\pi_{\zeta_g}}(s, a)) \log \pi_\theta(a \mid s) \right], \tag{18}$$

where $\beta$ is a hyper-parameter, $\exp_{clip}(\cdot)$ is the exponential function with a clipped range $(0, C]$, where $C$ is a positive number for numerical stability. We estimate the advantage using: $A^{\pi_{\zeta_g}}(s, a) \approx Q_\psi(s, a) - \mathbb{E}_{a \sim \pi_{\zeta_g}(\cdot|s)}[Q_\psi(s, a)]$, where we use the Q-function conditioned on the behaviour policy $\pi_b$ rather than $\pi_{\zeta_g}$ in order to reduce extrapolation error and computational complexity. Further, we employ the approach used in (Mao et al., 2023) to replace the expectation over $a \sim \pi_{\zeta_g}(\cdot \mid s)$ with the Q-value of the mean action of the Gaussian policy $\pi_{\zeta_g}$ to reduce computation.

Implementation details and the pseudocode can be found in Algorithm 1. The algorithm starts with training a MDN to model the behaviour policy, and then iteratively update the hyper Q-function $Q_\phi$, the value function $Q_\psi$ and the target policy $\pi_\theta$. The code has been open sourced [1].

## 6 Experimental Results

We aim to answer the following questions: (1) How does LOM compare to SOTA offline RL algorithms, particularly those handling multi-modality? (2) Can performance improvements be attributed to *learning on one mode*? (3) How does the number of Gaussian components affect LOM's performance?

### 6.1 LOM Achieves SOTA Performance in D4RL Benchmark

We evaluate LOM on three MuJoCo locomotion tasks from the D4RL benchmark (Fu et al., 2020): halfcheetah, hopper, and walker2d. Each environment contains five dataset types: (i) **medium** — 1M samples from a policy trained to approximately one-third of expert performance; (ii) **medium-replay** — the replay buffer of a policy trained to match the performance of the medium agent (0.2M for halfcheetah, 0.4M for hopper, 0.3M for walker2d); (iii) **medium-expert** — a 50-50 split of medium and expert data (just under 2M samples); (iv) **expert** — 1M samples from a fully trained SAC policy (Haarnoja et al., 2018); and (v) **full-replay** — 1M samples from the final replay buffer of an expert policy. Notably, the medium-replay and full-replay datasets are highly multi-modal, as states may have multiple action labels from different policies.

Table 1 shows the benchmark results of our method against SOTA offline algorithms: OneStep (Brandfonbrener et al., 2021), AWAC (Nair et al., 2021), CQL (Kumar et al., 2020), and TD3BC (Fujimoto & Gu, 2021). We also include four algorithms designed for handling multi-modality: LAPO (Chen et al., 2022), deterministic mixture policy optimisation (DMPO) (Osa et al., 2023),

---
[1]GitHub repository: https://github.com/MianchuWang/LOM

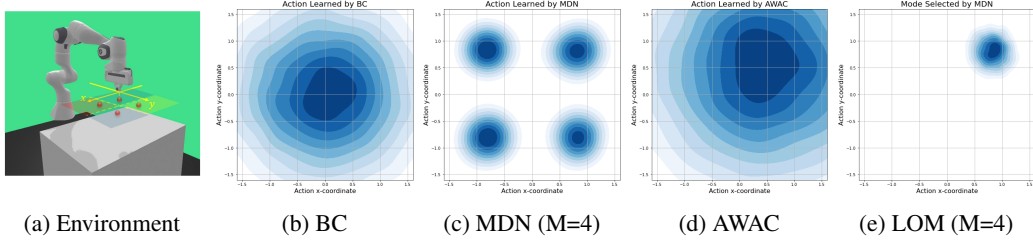

| (a) Environment | (b) BC | (c) MDN (M=4) | (d) AWAC | (e) LOM (M=4) |

Figure 2: Comparative study in the FetchReach task with highly multi-modal datasets. (a) The FetchReach robot is tasked with reaching one of four specified goals using an expert dataset. The robot arm receives a reward of 2 for reaching the goal in the first quadrant and 1 for reaching any of the other three goals. The dataset contains actions directed toward all four goals, with conflicting directions. (b) Action distribution learned by behaviour cloning using a unimodal Gaussian policy model. (c) Action distribution learned by MDN using a GMM policy model. (d) Action distribution learned by AWAC, which applies weighted imitation learning over the entire action distribution using a unimodal Gaussian policy. (e) Action distribution learned by LOM.

weighted conditioned GAN (WCGAN), and weighted conditioned VAE (WCVAE) (Wang et al., 2024b). The experimental results are shown in Table 1, where the baselines' results are mainly sourced from Mao et al. (2023) or reproduced using their official implementations. The results demonstrate that LOM outperforms the baseline algorithms in 12 out of 15 tasks. In tasks with multi-modal datasets, LOM surpasses all the baselines, and achieves a performance improvement ranging from 0.2% to 7.9% over the SOTA results.

## 6.2 LEARNING ON ONE MODE DRIVES IMPROVEMENTS

To explore the source of LOM's improvements, we design three tasks with highly multi-modal datasets: FetchReach, FetchPush, and FetchPickAndPlace. In FetchReach, as shown in Figure 2a, the objective is to control a robotic arm to reach one of four specified positions, symmetrically distributed on the $xy$-plane. The robot receives a reward of 2 for reaching the position in the first quadrant, 1 for reaching any of the other positions, and 0 otherwise. In FetchPush and FetchPickAndPlace, the robot is tasked with moving a cube to one of four positions on a desk, with the same reward structure. Each dataset contains $4 \times 10^6$ transitions collected by an expert policy. Given a state and four goals in different quadrants, the dataset contains four trajectories corresponding to each goal, making the datasets highly multi-modal.

We compare LOM with behaviour cloning (BC), AWAC, and MDN, introduced in Section 5.1. Specifically, BC and MDN imitate the behaviour policy using a unimodal Gaussian policy and a multi-modal Gaussian policy, respectively. AWAC applies weighted imitation learning to imitate the behaviour policy across all modes (Nair et al., 2021). LOM, however, distinguishes multiple behaviour modes using MDN and focuses on learning from the best mode. Table 2 shows that LOM outperforms the baseline algorithms by an average of 53%.

In Figure 2, we further investigate the behaviour of different policies. Given a state, the multi-modal dataset contains four actions, each targeting distinct goals in separate quadrants. Figure 2b shows that BC learns an averaged action distribution. Figure 2c demonstrates that MDN effectively reconstructs the action distribution, allowing LOM to select the highest-rewarding mode. Figure 2d shows that while AWAC can learn the best mode (in the first quadrant), it tends to generate OOD actions. In contrast, Figure 2e illustrates that LOM imitates actions from the best single mode, filtering out suboptimal actions and focusing on highly-rewarding ones. These experiments demonstrate that learning on one mode effectively improves policy performance by concentrating on the most rewarding actions.

Table 2: Average scores on the tasks with extremely multi-modal datasets.

|  | BC | MDN | AWAC | WCGAN | WCVAE | LOM |
|---|---|---|---|---|---|---|
| FetchReach | $14.3_{\pm 3.4}$ | $17.1_{\pm 2.3}$ | $32.4_{\pm 0.2}$ | $42.3_{\pm 3.1}$ | $40.8_{\pm 2.1}$ | $\mathbf{47.2}_{\pm 3.1}$ |
| FetchPush | $11.8_{\pm 0.9}$ | $18.9_{\pm 4.7}$ | $26.8_{\pm 1.4}$ | $41.2_{\pm 3.7}$ | $41.8_{\pm 2.4}$ | $\mathbf{44.3}_{\pm 2.7}$ |
| FetchPick | $3.6_{\pm 0.8}$ | $8.1_{\pm 1.3}$ | $22.7_{\pm 2.3}$ | $29.2_{\pm 2.8}$ | $30.5_{\pm 2.1}$ | $\mathbf{34.2}_{\pm 3.6}$ |

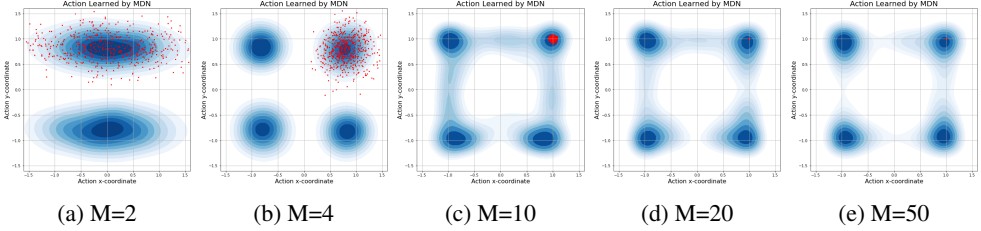

| (a) M=2 | (b) M=4 | (c) M=10 | (d) M=20 | (e) M=50 |

Figure 4: Action modes learned by MDN with varying numbers of mixtures. Red dots represent samples from the highest-reward mode. The original actions are clustered around $(1, 1)$, $(1, -1)$, $(-1, -1)$, and $(-1, 1)$ but do not extend beyond these points. In (d) and (e), the red dots collapse into a single point in the first quadrant due to the small standard deviation of the mode.

## 6.3 EFFECT OF THE NUMBER OF GAUSSIAN COMPONENTS

We analyse the influence of the number $M$ of Gaussian components in the LOM algorithm. The results, shown in Figure 4, are based on the multi-modal FetchReach environment. The figure illustrates the decomposition of action modes using blue contours, while red dots represent the actions imitated by LOM. When $M$ is too large (Figure 4d and 4e), the standard deviation of the imitated Gaussian mode becomes excessively small, leading to overly narrow the domains of action learning. Conversely, smaller values of $M$ results in modes with a large standard deviation (Figure 4a), causing LOM to capture actions outside the dataset and imitate OOD actions. This highlights the need to investigate LOM's sensitivity to this hyper-parameter.

Figure 3 compares LOM across $M = \{1, 5, 10, 15, 20\}$ in the medium-replay and full-replay datasets from the D4RL benchmark. The performance generally improves as the number of mixture components increases, particularly in the medium-replay dataset, where larger $M$ values lead to continuous performance gains. However, in the full-replay datasets, performance increases only up to $M = 10$. This indicates that while increasing the number of components benefits performance, there

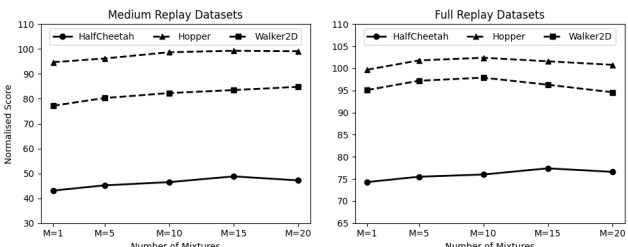

Figure 3: Normalised scores for varying numbers of Gaussian components in the medium replay and full replay datasets.

may be diminishing returns beyond a certain threshold. Our experiments suggest that $M$ primarily influences the domain of the imitated actions, and improvements can be achieved with a moderate, though not highly specific, value of $M$.

## 7 CONCLUSIONS AND DISCUSSIONS

In this paper, we introduced LOM, a novel offline RL method specifically designed to tackle the challenge of multi-modality in offline RL datasets. Unlike existing methods that aim to model the entire action distribution, LOM focuses on imitating the highest-rewarded action mode. Through extensive experiments, we demonstrated that LOM achieves SOTA performance, and we attributed these improvements to the central idea of *learning on one mode*, which simplifies the learning process while maintaining robust performance.

We observed that the hyper Q-function satisfies a Bellman-like equation, opening the possibility for learning it through Bellman updates. However, one challenge that emerges is the difficulty in accurately estimating the Bellman target for the hyper Q-function, as it requires computing the expectation over all possible hyper-actions, which may lead to extrapolation errors. Exploring a deterministic hyper-policy is a promising future direction to mitigate this issue, potentially simplifying the estimation process and further improving performance. Furthermore, the hyperparameter $M$ in LOM is intrinsically linked to the multi-modality present in the dataset. This parameter can be estimated using statistical techniques like bump hunting (Friedman & Fisher, 1999) and peak finding (Du et al., 2006), eliminating the need for fine-tuning. However, a key challenge lies in the fact that each state corresponds to a distinct multi-modal action distribution, making it computationally intensive to identify and count the modes for every individual state.

**Acknowledgments** We acknowledge support from a UKRI AI Turing Acceleration Fellowship (EPSRC EP/V024868/1).

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

## A   MOTIVATING EXAMPLE

To illustrate the challenge of capturing complex multi-modal action distributions, we present a toy example in Fig. 5. In this example, we generate a dataset where each state is associated with multiple valid actions, simulating the multi-modal nature of real-world offline RL datasets. We evaluate four models — Gaussian, Conditional Variational Auto-Encoder (CVAE) (Zhou et al., 2020; Chen et al., 2022), Conditional Generative Adversarial Networks (CGAN) (Yang et al., 2022b; Wang et al., 2024b), and Mixture Density Network (MDN) (Bishop, 1994) — on their ability to model the multi-modal action distribution.

Subfigures 5 (a-e) show the action distributions learned by each model. Specifically, we visualise how well each model captures distinct modes in the action space. The Gaussian model fails to separate modes, while the CVAE, CGAN, and MDN demonstrate stronger mode separation, with the MDN showing minimal mode overlap and the clearest boundaries between modes.

Additionally, we assess each model's ability to capture high-reward actions. Subfigures 5 (f-j) illustrate the models' performance in learning positively-rewarded multi-modal action distributions. Our results reveal that the MDN with ranked components closely adheres to in-distribution actions and achieves superior outcomes in terms of reward, compared to the other models.

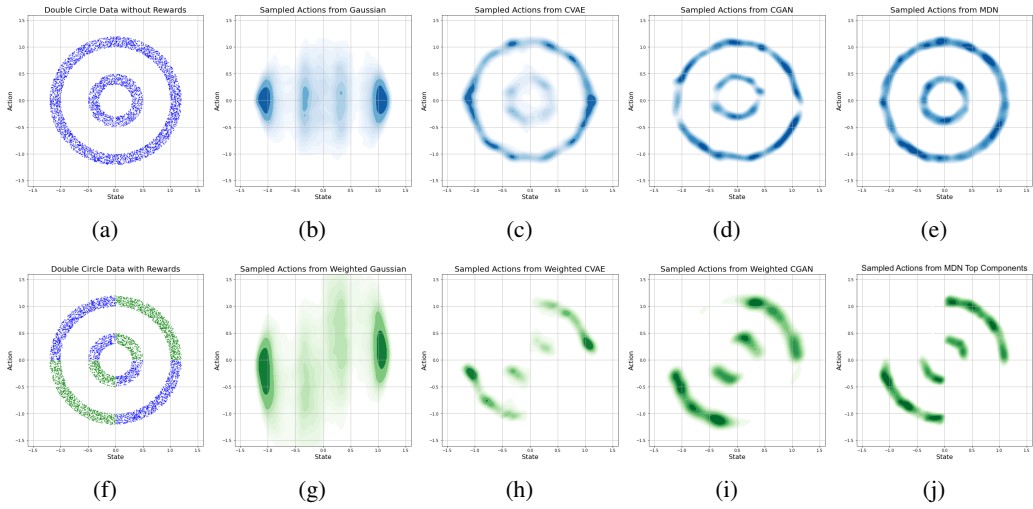

Figure 5: An example of modelling the multi-modal behaviour policy in a one-step MDP. The $x$-axis represents the state, and the $y$-axis represents the corresponding multi-modal actions. (a) shows the action distribution from the offline dataset. (b)-(e) illustrate the action distributions learned by a Gaussian model, a conditional VAE, a conditional GAN, and an MDN, respectively. (f) shows the action distribution of the offline dataset with rewards, where actions in the first and third quadrants receive a reward of 1, and others receive 0. (g)-(i) illustrate the action distributions learned by weighted Gaussian, weighted conditional VAE, and weighted conditional GAN models, respectively. (j) shows the action distribution learned by the top 10 of the 20 MDN components, ranked by a hyper Q-function.

## B   THEORETICAL RESULTS

### B.1   PROOF OF PROPOSITION 1

**Proposition 1** The hyper Q-function can be linked to the standard value function $Q^\pi(s, a)$ via:

$$Q_H^\zeta(s, u) = \mathbb{E}_{a \sim \phi^u(\cdot|s)} \left[ Q^{\pi_\zeta}(s, a) \right]. \tag{19}$$

*Proof:*

$$
\begin{aligned}
Q_H^{\zeta}(s,u) =& \mathbb{E}_{s_1,u_1,\cdots\sim\zeta}\left[\sum_{t=0}^{T}\gamma^t r_H(s_t,u_t)\mid s_0=s, u_0=u\right]\\
=& r_H(s,u) + \sum_{t=0}^{T-1}\sum_{s,u}p(s_{t+1}=s\mid\zeta)\zeta(u\mid s)\gamma^{t+1}r_H(s,u)\\
=& \sum_a\phi^u(a\mid s)r(s,a) + \sum_{t=0}^{T-1}\sum_s p(s_{t+1}=s\mid\zeta)\sum_u\zeta(u\mid s)\gamma^{t+1}\sum_a\phi^u(a\mid s)r(s,a)\\
=& \sum_a\phi^u(a\mid s)r(s,a) + \sum_{t=0}^{T-1}\sum_s p(s_{t+1}=s\mid\zeta)\sum_{u,a}\zeta(u\mid s)\phi^u(a\mid s)\gamma^{t+1}r(s,a)\\
=& \sum_a\phi^u(a\mid s)r(s,a) + \sum_{t=0}^{T-1}\sum_{s,a} p(s_{t+1}=s\mid\zeta)\pi_\zeta(a\mid s)\gamma^{t+1}r(s,a)\\
=& \sum_a\phi^u(a\mid s)\left[r(s,a) + \sum_{t=0}^{T-1}\sum_{s,a} p(s_{t+1}=s\mid\zeta)\pi_\zeta(a\mid s)\gamma^{t+1}r(s,a)\right]\\
=& \sum_a\phi^u(a\mid s)\left[r(s,a) + \sum_{t=0}^{T-1}\sum_{s,a} p(s_{t+1}=s\mid\pi_\zeta)\pi_\zeta(a\mid s)\gamma^{t+1}r(s,a)\right]\\
=& \mathbb{E}_{a\sim\phi^u(\cdot\mid s)}\left[Q^{\pi_\zeta}(s,a)\right].
\end{aligned}
\tag{20}
$$

## B.2    PROOF OF THEOREM 1

**Theorem 1.** The composite policy induced by the greedy hyper-policy improves upon the behaviour policy:
$$
V^{\pi_{\zeta_g}}(s) \geq V^{\pi_b}(s), \quad \forall s\in\mathcal{S}.
$$

*Proof.* Now, we prove that the improved policy $\pi_{\zeta_g}$ is uniformly as good as or better than the behaviour policy $\pi_b$.

$$
\begin{aligned}
V^{\pi_{\zeta_g}}(s_t) =& \mathbb{E}_{a_t\sim\pi_{\zeta_g}}[r(s_t,a_t)+\gamma\mathbb{E}_{s_{t+1},a_{t+1}\sim\pi_{\zeta_g}}[Q(s_{t+1},a_{t+1})]]\\
=& \mathbb{E}_{a_t\sim\pi_{\zeta_g}}[r(s_t,a_t)+\gamma\mathbb{E}_{s_{t+1},a_{t+1}\sim\pi_{\zeta_g}}[r(s_{t+1},a_{t+1})+\\
& \gamma\mathbb{E}_{s_{t+2},a_{t+2}\sim\pi_{\zeta_g}}[Q^{\pi_{\zeta_g}}(s_{t+2},a_{t+2})]]]\\
=& \mathbb{E}_{a_t\sim\pi_{\zeta_g}}[r(s_t,a_t)+\gamma\mathbb{E}_{s_{t+1},a_{t+1}\sim\pi_{\zeta_g}}[r(s_{t+1},a_{t+1})+\cdots+\\
& \gamma\mathbb{E}_{s_{t+H-1},a_{t+H-1}\sim\pi_{\zeta_g}}[r(s_{t+H-1},a_{t+H-1})+\gamma\mathbb{E}_{s_{t+H},a_{t+H}\sim\pi_{\zeta_g}}[r(s_{t+H},a_{t+H})]]]\\
\geq& \mathbb{E}_{a_t\sim\pi_{\zeta_g}}[r(s_t,a_t)+\gamma\mathbb{E}_{s_{t+1},a_{t+1}\sim\pi_{\zeta_g}}[r(s_{t+1},a_{t+1})+\cdots+\\
& \gamma\mathbb{E}_{s_{t+H-1},a_{t+H-1}\sim\pi_{\zeta_g}}[r(s_{t+H-1},a_{t+H-1})+\gamma\mathbb{E}_{s_{t+H},a_{t+H}\sim\pi_b}[r(s_{t+H},a_{t+H})]]]\\
=& \mathbb{E}_{a_t\sim\pi_{\zeta_g}}[r(s_t,a_t)+\gamma\mathbb{E}_{s_{t+1},a_{t+1}\sim\pi_{\zeta_g}}[r(s_{t+1},a_{t+1})+\cdots+\\
& \gamma\mathbb{E}_{s_{t+H-1},a_{t+H-1}\sim\pi_{\zeta_g}}[Q^{\pi_b}(s_{t+H-1},a_{t+H-1})]]]\\
\geq& \mathbb{E}_{a_t\sim\pi_{\zeta_g}}[r(s_t,a_t)+\gamma\mathbb{E}_{s_{t+1},a_{t+1}\sim\pi_{\zeta_g}}[r(s_{t+1},a_{t+1})+\cdots+\\
& \gamma\mathbb{E}_{s_{t+H-1},a_{t+H-1}\sim\pi_b}[Q^{\pi_b}(s_{t+H-1},a_{t+H-1})]]]\\
\geq& \cdots\\
\geq& \mathbb{E}_{a_t\sim\pi_{\zeta_g}}[r(s_t,a_t)+\gamma\mathbb{E}_{s_{t+1},a_{t+1}\sim\tilde\pi_b}[Q^{\pi_b}(s_{t+1},a_{t+1})]]]\\
\geq& \mathbb{E}_{a_t\sim\pi_b}[r(s_t,a_t)+\gamma\mathbb{E}_{s_{t+1},a_{t+1}\sim\tilde\pi_b}[Q^{\pi_b}(s_{t+1},a_{t+1})]]]\\
=& V^{\pi_b}(s_t)
\end{aligned}
$$

(21)

(22)

The derivation from Eq. 21 to Eq. 22 is based on:

$$
\begin{aligned}
&\mathbb{E}_{s,a\sim\pi_{\zeta_g}}[Q^{\pi_b}(s,a)] \\
&= \mathbb{E}_{s,u\sim\zeta_g}\mathbb{E}_{a\sim\phi^u(\cdot|s)}[Q^{\pi_b}(s,a)] \\
&= \mathbb{E}_s\mathbb{E}_{u\sim\zeta_g}[Q_H^{\zeta_b}(s,u)] \\
&\geq \mathbb{E}_s\mathbb{E}_{u\sim\zeta_b}[Q_H^{\zeta_b}(s,u)] \\
&= \mathbb{E}_{s,u\sim\zeta_b}\mathbb{E}_{a\sim\phi^u(\cdot|s)}[Q^{\pi_b}(s,a)] \\
&= \mathbb{E}_{s,a\sim\pi_b}[Q^{\pi_b}(s,a)]
\end{aligned}
\tag{23}
$$

$\square$

### B.3 PROOF OF THEOREM 2

**Theorem 2** LOM learns a policy $\pi_L$, which is uniformly as good as or better than the improved policy $\pi_{\zeta_g}$ and the behaviour policy $\pi_b$. That is,

$$
\forall s \in \mathcal{S}, V^{\pi_L}(s) \geq V^{\pi_{\zeta_g}}(s) \geq V^{\pi_b}(s).
\tag{24}
$$

*Proof.* The proof is structured in two parts. First, we show that the LOM-learned policy $\pi_L$ is at least as good as the improved policy $\pi_{\zeta_g}$; specifically, for all states $s \in \mathcal{S}$, we establish that $V^{\pi_L}(s) \geq V^{\pi_{\zeta_g}}(s)$. Second, we prove that the improved policy $\pi_{\zeta_g}$ performs no worse than the behaviour policy $\pi_b$, i.e., $V^{\pi_{\zeta_g}}(s) \geq V^{\pi_b}(s)$ for all $s \in \mathcal{S}$.

As established in Wang et al. (2018), the following lemma provides a sufficient condition for proving that one policy is no worse than another. We restate this result as the following lemma:

**Lemma (from Wang et al. (2018)):** Suppose two policies $\pi$ and $\tilde{\pi}$ satisfy:

$$
g(\tilde{\pi}(a \mid s)) = g(\pi(a \mid s)) + h(s, A^{\pi}(s,a)),
\tag{25}
$$

where $g(\cdot)$ is a monotonically increasing function, and $h(s, \cdot)$ is monotonically increasing for any fixed $s$. Then we have:

$$
V^{\tilde{\pi}}(s) \geq V^{\pi}(s), \quad \forall s \in \mathcal{S}.
\tag{26}
$$

This means that $\tilde{\pi}$ is uniformly as good as or better than $\pi$.

Since the LOM-learned policy imitates the optimal policy $\pi^*$, we have:

$$
\pi^*(a \mid s) = \frac{1}{Z(s)}\pi_{\zeta_g}(a \mid s)\exp\left(\frac{1}{\beta}A^{\pi_{\zeta_g}}(s,a)\right),
\tag{27}
$$

which gives:

$$
\log\pi_L(a \mid s) = \log\frac{1}{Z(s)} + \log\pi_{\zeta_g}(a \mid s) + \frac{1}{\beta}A^{\pi_{\zeta_g}}(s,a),
\tag{28}
$$

where $\log(\cdot)$ is a monotonically increasing function and $\log\frac{1}{Z(s)}$ is a constant. Therefore, $\pi_L$ is uniformly as good as or better than the improved policy $\pi_{\zeta_g}$. $\square$

### B.4 PROOF OF THEOREM 3

**Theorem 3** The LOM algorithm learns a policy $\pi_L$, whose improvement upon the composite policy induced by the greedy hyper-policy $\pi_{\zeta_g}$ is bounded: for all $s \in \mathcal{S}$:

$$
V^{\pi_L}(s) - V^{\pi_{\zeta_g}}(s) \geq \frac{1}{1-\gamma}\hat{\eta}(\pi_L) - \frac{A_{\max}}{1-\gamma}\sqrt{\frac{1}{2}D_{KL}(d_{\pi_L}||d_{\pi_{\zeta_g}})},
\tag{29}
$$

where $A_{\max} = \max_{s,a}|A^{\pi_{\zeta_g}}(s,a)|$. The proof is provided in Appendix B.4.

*Proof.* Following performance difference lemma (Kakade & Langford, 2002), we have

$$
V^{\pi_L}(s) - V^{\pi_{\zeta_g}}(s) = \frac{1}{1-\gamma}\mathbb{E}_{s\sim d_{\pi_L}, a\sim\pi_L}\left[A^{\pi_{\zeta_g}}(s,a)\right].
\tag{30}
$$

Our goal is to express the right-hand side in terms of $\hat{\eta}(\pi_L)$, which uses $d_{\pi_g}$ rather than $d_{\pi_L}$. To do this, we define the discrepancy between $d_{\pi_L}$ and $d_{\pi_{\zeta_g}}$ as:

$$\Delta(s) = d_{\pi_L}(s) - d_{\pi_{\zeta_g}}(s). \tag{31}$$

We can decompose the expectation over $d_{\pi_L}$ as:

$$\mathbb{E}_{s \sim d_{\pi_L}}[f(s)] = \mathbb{E}_{s \sim d_{\pi_{\zeta_g}}}[f(s)] + \mathbb{E}_{s \sim \Delta}[f(s)], \tag{32}$$

where $\mathbb{E}_{s \sim \Delta}[f(s)] = \sum_s \Delta(s) f(s)$.

Applying this to the expression:

$$V^{\pi_L}(s) - V^{\pi_{\zeta_g}}(s) = \frac{1}{1-\gamma} \left( \mathbb{E}_{s \sim d_{\pi_{\zeta_g}}, a \sim \pi_L} \left[ A^{\pi_{\zeta_g}}(s,a) \right] + \mathbb{E}_{s \sim \Delta, a \sim \pi_L} \left[ A^{\pi_{\zeta_g}}(s,a) \right] \right). \tag{33}$$

The first term is exactly $\hat{\eta}(\pi_L)$ scaled by $\frac{1}{1-\gamma}$. We need to bound the error term:

$$\epsilon = \frac{1}{1-\gamma} \mathbb{E}_{s \sim \Delta, a \sim \pi_L} \left[ A^{\pi_{\zeta_g}}(s,a) \right] \tag{34}$$

Assuming the advantage function $A^{\pi_{\zeta_g}}(s,a)$ is bounded:

$$|A^{\pi_{\zeta_g}}(s,a)| \leq A_{\max}. \tag{35}$$

We can then bound $\epsilon$ by

$$|\epsilon| \leq \frac{A_{\max}}{1-\gamma} \sum_s |\Delta(s)| = \frac{A_{\max}}{1-\gamma} ||d_{\pi_L} - d_{\pi_{\zeta_g}}||_1 = \frac{A_{\max}}{1-\gamma} D_{TV}(d_{\pi_L}||d_{\pi_{\zeta_g}}) \tag{36}$$

By applying Pinsker's inequity, we have

$$D_{TV}(d_{\pi_L}||d_{\pi_{\zeta_g}}) \leq \sqrt{\frac{1}{2} D_{KL}(d_{\pi_L}||d_{\pi_{\zeta_g}})} \tag{37}$$

Finally, we have

$$V^{\pi_L}(s) \geq V^{\pi_{\zeta_g}}(s) + \frac{1}{1-\gamma} \hat{\eta}(\pi_L) - \frac{A_{\max}}{1-\gamma} \sqrt{\frac{1}{2} D_{KL}(d_{\pi_L}||d_{\pi_{\zeta_g}})} \tag{38}$$

$\square$

## C  FURTHER EXPERIMENTAL DETAILS

### C.1  BASELINES

In the experiments discussed in Section 6, we compare LOM with other multi-modal modelling approaches, such as WCGAN and WCVAE, in capturing complex action distributions in offline reinforcement learning. These models are evaluated on their ability to reconstruct and learn from the multi-modal behaviour policy in the FetchReach, FetchPush, and FetchPickAndPlace tasks, and across various D4RL benchmarks.

**Weighted Conditional GAN (WCGAN):**  The WCGAN learns a discriminator $D$ and a generator $\pi$ to optimise the following adversarial objectives:

$$\max_D \min_\pi \quad \mathbb{E}_{(s,a) \sim \mathcal{D}} [w(s,a) \log D(s,a)] + \mathbb{E}_{s \sim \mathcal{D}, a' \sim \pi(\cdot|s)} [\log(1 - D(s,a'))], \tag{39}$$

where $w(s,a) = \exp_{clip} \left( \frac{1}{\beta} A^{\pi_b}(s,a) \right)$. Here, $A^{\pi_b}$ represents the advantage of the behaviour policy $\pi_b$. The WCGAN aims to generate action distributions that are weighted by this advantage, prioritising actions that yield higher returns.

**Weighted Conditional VAE (WCVAE):**  The WCVAE learns a conditional variational autoencoder with an encoder $E$ and decoder $D$. The objective is to optimize:

$$\max_{E,D} \quad \mathbb{E}_{(s,a) \sim \mathcal{D}, z \sim E(s,a)} [w(s,a) \log D(a \mid z, s) - w(s,a) D_{KL}(E(z \mid s,a) \parallel p(z \mid s))], \tag{40}$$

where $p(z \mid s)$ is the unit Gaussian prior distribution of the latent variable conditioned on $s$. The weighting function $w(s,a)$ encourages the WCVAE to focus on high-reward actions.

### C.2  HYPERPARAMETERS

The hyperparameters used in LOM were largely the same for all of the experiments reported. In Table 3, we give a list and description of them, as well as their default values. The hyperparameters $M$ used in Table 1 are selected from $\{2, 5, 10, 15, 20\}$.

| Symbol in paper | Description | Default values |
|:---:|:---|:---:|
| $M$ | Number of Gaussian components | Please check Figure 3 |
| $\beta$ | Advantage weights | 5 |
| $C$ | Weight clips | 50 |
| $\rho$ | Ployak averaging coefficient | 0.995 |
| update_delay | Target update delay | 2 |
| $\pi_\rho$ | Gaussian mixture model | [state_dim, 512, 512, 2 * ac_dim + $M$] |
| $\pi_\theta$ | Policy learned by LOM | [state_dim, 512, 512, ac_dim] |
| $Q_\psi$ | Q-network | [state_dim + ac_dim, 512, 512, 1] |
| $Q_\phi$ | Behaviour hyper Q-function | [state_dim + $M$, 512, 512, $M$] |

Table 3: Hyperparameters used in the experiments.

