# OpenReview forum: "Learning on One Mode: Addressing Multi-modality in Offline Reinforcement Learning"
_ICLR.cc/2025/Conference — ICLR 2025 Poster_

### Official Review · Reviewer_XhoW · 2024-10-29

**Soundness:** 2
**Presentation:** 2
**Contribution:** 3
**Rating:** 6
**Confidence:** 3

**Summary:**

This paper presents an offline reinforcement learning algorithm that accounts for the multimodal distribution of actions within a given dataset. In the proposed algorithm, the behavior policy is modeled as a Gaussian mixture policy, and the mode yielding the highest expected return is identified based on an approximated Q-function associated with the behavior policy. The policy is then learned using an advantage-weighted actor-critic approach, guided by actions generated by the mode of the behavior policy that maximizes the expected return.

The proposed method was evaluated on standard D4RL tasks and custom tasks based on Fetch tasks with multimodal datasets. Experimental results show that the proposed method outperformed baseline methods.

**Strengths:**

- The idea is interesting and reasonable. While existing methods often focus on enhancing the flexibility of the policy model, the proposed approach aims to extract a unimodal distribution from a multimodal dataset.

- In the experiments, the proposed method outperformed the baseline methods.

**Weaknesses:**

- The comparison with related methods is insufficient. While the proposed approach is interesting for its aim to extract a unimodal action distribution from a given dataset, several methods enhance policy expressiveness to handle multimodal action distributions. Using a Gaussian mixture model is a straightforward and intuitive approach. A recent study explores Gaussian mixture policies and mixtures of deterministic policies based on AWAC (https://openreview.net/forum?id=zkRCp4RmAF). Additionally, diffusion models offer a promising approach (https://openreview.net/forum?id=AHvFDPi-FA). Comparisons with these methods should be included in the experiments.

- Hyperparameters used in the experiments are not provided. Please provide a table or appendix section detailing the hyperparameters used for each experiment, including any tuning process.

**Questions:**

- Why are LAPO, WCGAN, and WCVAE absent from the experiments with extremely multimodal datasets? These methods should be compared to evaluate their capability in handling multimodal action distributions.

- Is it possible to include Diffusion QL and AWAC with a Gaussian policy for comparison? These methods should also be included in the experiments with extremely multimodal datasets.

- In the attached code, I noticed that the loss function for the GMM might be incorrect. The provided code shows the loss as $\mathcal{L} = \sum_i ( \log w_i + \log p_i$ ) where $p_i$ is a Gaussian component.
However, the likelihood in a GMM is given by $\sum w_i p_i $ and its log likelihood is $\log ( \sum w_i p_i )$, which differs from $ \sum_i ( \log w_i + \log p_i$ )$.  Was this modification made intentionally or by mistake? If it was intentional, please provide the rationale.

- How sensitive is the proposed algorithm to the number of Gaussian components? Please provide an ablation study or sensitivity analysis showing how performance changes with different numbers of Gaussian components across various tasks or datasets.

=== comments after the rebuttal ===

The concerns raised in the initial review were appropriately addressed in the rebuttal.

---

> ### Author Response · Authors · 2024-11-25
>
> Thank you sincerely for the effort and time you spent reviewing our manuscript. We have taken care to address each point you raised, making revisions highlighted in blue within the revised paper. We believe these adjustments will effectively alleviate your concerns.
>
> **W1.** We agree that several methods enhance policy expressiveness to handle multi-modal action distributions; however, our approach is fundamentally different. Instead of aiming to model or express the entire multi-modal distribution, we explicitly decompose the behaviour policy into its constituent modes and actively select the best mode to generate actions at each state.
>
> The methods mentioned by the reviewer focus on capturing multi-modality for expressive policies, while our goal is to simplify the learning process by isolating and leveraging the most relevant mode for decision-making. As such, these methods are not in direct competition with our approach but could provide additional baselines. Some of the strong baselines had already been included in the original submission; specifically, our original Table 1 included LAPO, WCGAN, and WCVAE. Now, in Table 1, we have also included DMPO as you suggested.
>
>
> **W2.** We have added the hyperparameter details to the Appendix for clarity and completeness.
>
> **Q1.** In the original Section 6.2, we focused on demonstrating that learning on one mode improves upon behaviour cloning (BC and MDN) and weighted imitation learning (AWAC and LOM). To address the reviewer's request, in the revised version, we have added WCGAN and WCVAE as baselines.
>
> The results show that both methods handle multi-modality better than a single Gaussian learned by AWAC but still underperform compared to LOM. This is because WCGAN and WCVAE cannot entirely eliminate actions from sub-optimal modes, which LOM addresses explicitly.
>
> **Q2.** The benchmark involves tasks with highly multi-modal action distributions, where the policy must select from diverse behaviours to achieve specific predefined goals. While we acknowledge the relevance of LAPO, Diffusion QL, and DMPO, and plan to include these comparisons in future revisions, we believe that the current strong baselines already cover a broad range of different approaches. Our empirical evidence provides strong support for the effectiveness of the idea we propose. The comparisons, though interesting, are unlikely to change the fundamental claim of our paper—that focusing on mode selection has the potential to improve policy performance in multi-modal scenarios.
>
> **Q3.** Thanks for spotting this! The log-likelihood in a GMM is $\log \sum_i w_i p_i$, where $w_i$ is the mixing coefficient and $p_i$ is the Gaussian component. In the code, we maximised $\sum_i (\log w_i + \log p_i)$ but this was a mistake. Fortunately, we found that the two loss functions produce comparable results and do not affect our overall conclusions. We have updated all the experimental results accordingly in the revised manuscript.
>
> **Q4.** In the original paper, we have provided an ablation study in Section 6.3, where we evaluated LOM with different numbers of Gaussian components $M = \\{1, 5, 10, 15, 20\\}$ and analysed the effects of these choices. The study demonstrated that $M$ influences the domain of the imitated actions, with improvements achieved at moderate values of $M$. This highlights the importance of balancing expressiveness and generalisation when selecting $M$. In the reply to reviewer N1vf, we have also added additional experiments showing the performance when $M$ is tuned fully offline using cross-validation to optimise the goodness-of-fit metric for the mixture of Gaussians. This demonstrates a practical approach to parameter selection in a purely offline setting.
>
> Please note that optimal tuning of offline RL parameters in a fully offline manner remains an open problem. LOM has only a main parameter to tune, making it relatively simple compared to other methods that involve a larger number of hyperparameters.
>
> Finally, we thank you again for raising important questions about LOM. Have we adequately addressed the main concerns? Please feel free to let us know if there are additional concerns or questions.

---

> ### Author Response · Authors · 2024-11-30
>
> Hi Reviewer XhoW,
>
> Thank you for your valuable suggestions on our manuscript. We have incorporated your feedback and made improvements to the paper. We kindly ask if you would consider revising the review score. Your further input would also be greatly appreciated.

---

> > ### Comment · Reviewer_XhoW · 2024-12-01
> > **Thank you for the response**
> >
> > Thank you for addressing the questions and requests from the initial review. Most of my concerns have been resolved.
> >
> > However, I would like to ask one point: the authors mentioned that the loss function of the GMM was incorrect. Does the performance of the proposed method in the current manuscript reflect the results after correcting the GMM loss function? I would like to confirm that the correction of the GMM loss function did not significantly impact the performance of the proposed method.

---

> > > ### Author Response · Authors · 2024-12-02
> > >
> > > Thank you for your follow-up question. Regarding your query about the GMM loss function, we confirm that the performance results presented in the revised manuscript reflect the corrected log-likelihood formulation for the GMM. After identifying the issue, we re-ran the experiments with the correct log-likelihood formulation, and we found that the practical difference between the two loss functions is minor. The anonymous repository has been updated accordingly. We sincerely appreciate your valuable contribution and thoughtful feedback on our work.

---

### Official Review · Reviewer_HzTN · 2024-10-31

**Soundness:** 3
**Presentation:** 3
**Contribution:** 3
**Rating:** 8
**Confidence:** 3

**Summary:**

This paper addresses the multi-modality problem in offline reinforcement learning (RL). It proposes to first perform model selection and then further improve the greedy hyper policy that captures the best mode.

**Strengths:**

I enjoy reading this paper. I think its key contribution is learning on one mode. By doing this, the KL-divergence term in offline policy improvement would not burden the policy to cover the entire support distribution generated by the behavior policy.

The key problem lies in the fact that, in the behavior data, the action distribution has some clear modes that should be represented by discrete values and some continuously distributed probability mass within each mode. The paper proposes dealing with these two aspects using different approaches. For the first aspect, it involves selecting between different modes to ensure the learned policy remains within the offline data distribution. For the second aspect, it involves using offline policy improvement with KL-divergence to constrain the continuous action distribution.

**Weaknesses:**

No clear weaknesses.

**Questions:**

I am curious about the theoretical aspect. Clearly, this paper employs a two-step improvement process. Empirically, it seems reasonable that this approach would perform better than doing just the second step without the mode selection. However, I am wondering if there is any theoretical analysis comparing the two-step improvement with only performing offline policy improvement in the second step. Which approach do you think would yield better results theoretically?

---

> ### Author Response · Authors · 2024-11-26
>
> Thank you sincerely for the effort and time you have spent reviewing our manuscript. We have carefully analysed and discussed your valuable question.
>
> We believe that LOM can yield better results theoretically under certain conditions. Let us denote the policy that performs only offline policy improvement in the second step (without mode selection) as $\pi_2$. Since $\pi_2$ averages over conflicting and suboptimal modes, its expected value $V^{\pi_2}(s)$ may be lower than the Q-values of high-performing actions selected by $\pi_L$. We assume the following condition: for any states $s$ from the initial state distribution, we have:
> \begin{equation}
>     E_{a \sim \pi_L(\cdot \mid s)} [Q^{\pi_2}(s, a)] \geq V^{\pi_2}(s).
> \end{equation}
> This implies:
> \begin{equation}
>     E_{a \sim \pi_L(\cdot \mid s)} [A^{\pi_2}(s, a)] \geq 0,
> \end{equation}
> where $A^{\pi_2}(s, a) = Q^{\pi_2}(s, a) - V^{\pi_2}(s)$ is the advantage function under policy $\pi_2$. With the expected advantage being non-negative, the performance difference between $\pi_L$ and $\pi_2$ becomes:
> \begin{equation}
>     V^{\pi_L}(s) - V^{\pi_2}(s) = \frac{1}{1 - \gamma} E_{s' \sim d_{\pi_L},\ a \sim \pi_L(\cdot \mid s')} [A^{\pi_2}(s', a)] \geq 0.
> \end{equation}
> This result follows from the performance difference lemma (Kakade and Langford., 2002; Schulman et al., 2015).
>
> Therefore, under the condition, LOM can produce a better policy than the one learned from performing only offline policy improvement without mode selection. We recognise that this is an interesting problem, and we will investigate it further—specifically, we will determine the exact conditions under which the policy improvement holds.
>
> We would like to thank you again for raising the important question about LOM. Please feel free to let us know if you have any additional concerns or questions.
>
> **References**
>
> Kakade, S., \& Langford, J. (2002). Approximately Optimal Approximate Reinforcement Learning. ICML.
>
> Schulman, J., Levine, S., Abbeel, P., Jordan, M., \& Moritz, P. (2015). Trust Region Policy Optimization. ICML.

---

### Official Review · Reviewer_Eg5z · 2024-11-02

**Soundness:** 2
**Presentation:** 3
**Contribution:** 2
**Rating:** 6
**Confidence:** 2

**Summary:**

The paper presents LOM, an offline imitation learning approach designed to address the multi-modality of batch datasets. The method employs a Gaussian mixture model to capture the behavior policy, selecting the optimal mode based on expected returns. Subsequently, it estimates the advantage by leveraging information from the selected mode and incorporates this advantage into a weighted imitation learning objective using an exponential weighting scheme. The approach demonstrates competitive performance against baselines on the D4RL benchmark.

**Strengths:**

1. The paper is well-written and easy to follow.

2. It effectively leverages the multimodal nature of the data to design a practical method.

**Weaknesses:**

1. While the method appears broadly applicable and versatile, it closely resembles existing techniques, such as weighted advantage imitation learning and multi-modal policy networks. The approach mainly combines established methods, making the contribution appear incremental.
2. Furthermore, although the author analyzes how the number of mixture components impacts the results, the experimental findings require additional clarification. For instance, in Section 6.3, the statement “When M is too large, it results in overly narrow domains of action learning, which may exclude potentially useful actions from the dataset” would benefit from a more thorough explanation, including potential reasons behind this effect. Without such elaboration, the analysis may feel incomplete.

**Questions:**

1. In mode selection, the greedy policy is applied. Is this the optimal strategy for choosing the mode? I wonder if a more conservative approach might yield better results. Could the author elaborate more about this choice?

---

> ### Author Response · Authors · 2024-11-21
>
> We are grateful for your time in evaluating our manuscript. The manuscript has been accordingly updated in blue to reflect these changes. We hope that our responses will effectively resolve your concerns.
>
> **W1.** We appreciate your observation regarding the use of established techniques. While LOM employs tools like GMMs and weighted imitation learning, its novelty lies in reframing the problem as a mode-selection problem. To the best of our knowledge—and as acknowledged by the reviewers—this approach is original.
>
> Unlike methods that model the entire multi-modal action distribution (e.g., GANs or VAEs) or assume unimodality, LOM focuses exclusively on the highest-reward mode. This reframing enables a simpler, more efficient policy learning process that avoids unnecessary complexity while maintaining robust theoretical guarantees (Theorems 1-3). By selecting and imitating a single, most promising mode, LOM addresses multi-modality in a way that is both practical and novel. More detailed comparisons to existing work are presented in the Multi-modality in offline RL, Section 2.
>
>
> **W2.** In Figure 4, when we increase the number of Gaussian components, the standard deviation of the highest-reward mode becomes smaller and smaller (the area of the red dots are gradually collapse into a single point). Since the policy $\pi_L$ is learned by imitating the actions from the highest-rewarded mode (i.e., the red dots), we say increasing the number of Gaussian components narrows domains of action learning. It is as narrow as a dot, reducing the potentially useful actions. You may zoom in the Figure to check the highest-reward mode in Figure 4.d and Figure 4.e.
>
> **Q1.** The conservative nature of the hyper Q-function and the weighted imitation learning method make us use a greedy policy safely. The hyper Q-function conditioned on the behaviour policy inherently regularizes action evaluation, maintaining a conservative stance (Kumar et al., 2020). The imitated action mode is derived from a fixed dataset, so the learned policy is constrained towards the behaviour policy in terms of KL-divergence (Equation 9).
>
> Following the suggestion, we modify the definition in Equation 7 with $\zeta_g(s) = \arg \max_{u \in \mathcal{A}} \alpha^u(s)$. Here $\zeta_g$ is a conservative hyper-policy which select the mode that the behaviour policy most likely to take. We report the experiment results in the following table. It shows the conservative mode selection method is worse than a greedy selection, indicating that overly constraining to the behaviour policy limits policy improvement.
>
> | **Benchmark**              | **LOM**             | **Conservative LOM**          |
> |----------------------------|---------------------|--------------------------------|
> | Halfcheetah-medium-replay  | $48.8_{\pm 0.7}$    | $46.3_{\pm 0.9}$               |
> | Hopper-medium-replay       | $99.2_{\pm 1.1}$    | $94.2_{\pm 1.0}$               |
> | Walker2d-medium-replay     | $84.8_{\pm 1.0}$    | $82.3_{\pm 1.4}$               |
> | HalfCheetah-full-replay    | $76.6_{\pm 1.2}$    | $76.3_{\pm 2.4}$               |
> | Hopper-full-replay         | $102.0_{\pm 2.7}$   | $100.8_{\pm 1.7}$              |
> | Walker2d-full-replay       | $97.9_{\pm 0.9}$    | $97.4_{\pm 0.5}$               |
>
> We would like to thank you again for raising these questions about LOM. Have we sufficiently addressed the main concerns? Please feel free to let us know if there are additional concerns or questions.
>
> **Reference**
>
> Kumar, A., Zhou, A., Tucker, G., & Levine, S. (2020). Conservative Q-Learning for Offline Reinforcement Learning. NeurIPS.

---

> > ### Comment · Reviewer_Eg5z · 2024-11-24
> >
> > Thank you for your clarification. It addresses my concern, and I will adjust the score accordingly.

---

> > > ### Author Response · Authors · 2024-11-26
> > >
> > > Thank you for taking the time to evaluate our paper. We truly appreciate your support and recognition.

---

### Official Review · Reviewer_N1vf · 2024-11-04

**Soundness:** 3
**Presentation:** 3
**Contribution:** 3
**Rating:** 6
**Confidence:** 3

**Summary:**

This paper introduces Weighted Imitation Learning on One Mode (LOM), a novel approach for offline reinforcement learning that addresses the challenges of multi-modal action distributions by focusing on the single mode with the highest expected return. Using a Gaussian mixture model to identify modes and a hyper Q-function for mode selection, LOM significantly improves performance in multi-modal environments, outperforming other methods on standard benchmarks.

**Strengths:**

1. This paper is well-organized and easy to follow.
2. The paper proposes an innovative approach to handling multi-modal action distributions in offline reinforcement learning, which provides valuable insights for research of muti-modality in RL, by focusing on the most promising action mode and using a Gaussian mixture model to identify modes.
3. This paper presents extensive experiments validating the effectiveness of LOM across diverse benchmarks. The results demonstrate that LOM outperforms state-of-the-art offline RL algorithms in multi-modal settings.

**Weaknesses:**

1. For the theoretical perspective, the result that $V^{\pi^L}(s) \geq V^{\pi^{\zeta_g}}(s) \geq V^{\pi^b}(s)$ is relatively weak, Is it possible to provide a lower bound for the gap between the value functions, or could the author explain the difficulties in doing so?
2. The effectiveness of LOM depends on the choice of the number of Gaussian components (M) in the Gaussian Mixture Model. Have the author tried any automatic model selection techniques for GMMs, such as the Bayesian Information Criterion or cross-validation, and how these might be applied to LOM?

**Questions:**

See weakness.

---

> ### Author Response · Authors · 2024-11-21
> **Reply to W1**
>
> We sincerely appreciate the time and effort you dedicated to reviewing our paper. We have addressed each of these points individually and revised the paper in blue accordingly. We hope that our responses will effectively resolve your concerns.
>
> **W1.** Our original Theorem 2 demonstrates that the proposed algorithm learns a policy $\pi_L$ that is at least as good as both the composite policy induced by the greedy hyper-policy $\pi_{\zeta_g}$ and the behaviour policy $\pi_b$. Although we did not provide lower bounds in our original submission, we believe that this result already adds theoretical support to the strong empirical performance.
>
> We have taken your comment into consideration, and have attempted to derive such bounds. This further analysis has been included in the manuscript as Theorem 3, and we summarise these additions in what follows.
>
> There are two inequalities in the theorem: $V^{\pi_L}(s) \geq V^{\pi_{\zeta_g}}(s)$ and $V^{\pi_{\zeta_g}}(s) \geq V^{\pi_b}(s)$. We have been able to derive a lower bound for the first inequality and have added it to the revised manuscript as Theorem 3. This result states that, $\forall s \in \mathcal{S}$,
> \begin{equation}
> V^{\pi_L}(s) - V^{\pi_{\zeta_g}}(s) \geq \frac{1}{1-\gamma} \hat{\eta}(\pi_L) - \frac{A_{\max}}{1-\gamma} \sqrt{\frac{1}{2} D_{KL}(d_{\pi_L} || d_{\pi_{\zeta_g}})},
> \end{equation}
> where $A_{\max} = \max_{s, a} |A^{\pi_{\zeta_g}}(s, a)|$, $\hat{\eta}(\cdot)$ is the approximated expected improvement (defined in Section 4.5), $\pi_L$ is the policy optimising the LOM objective (Equation 11). The equality holds when $\pi_L(a \mid s) = \pi_{\zeta_g}(a \mid s)$ for all $s \in \mathcal{S}$, as the policy improvement $\hat{\eta}(\pi_L)$ equals zero and the KL-divergence $D_{KL}(d_{\pi_L} || d_{\pi_{\zeta_g}})$ is also zero. The bound shows that the performance of the learned policy $\pi_L$ is guaranteed with the expected performance improvement quantified by the advantage and reduced by a penalty term proportional to the divergence in their state visitation distributions—emphasizing that maximizing advantage while minimizing divergence leads to better performance.
>
>
> To establish a lower bound for the second inequality, we need to relate $V^{\pi_{\zeta_g}}$ and $V^{\pi_b}$. This can be achieved by constructing the following equation:
> \begin{equation}
>     V^{\pi_b}(s_t) = \sum_a \pi_b(a_t \mid s_t) [r(s_t, a_t) + \gamma E_{s_{t+1} \sim \mathcal{P}(\cdot \mid s_t, a_t)} [V^{\pi_b}(s_{t+1})]] = \sum^M_{i=1} \alpha^i(s_t) \sum_a \phi_i(a_t \mid s_t) [r(s_t, a_t) + \gamma E_{s_{t+1} \sim \mathcal{P}(\cdot \mid s_t, a_t)} [V^{\pi_b}(s_{t+1})]]
> \end{equation}
>
> Here we define $\alpha_{\max}=\max_{s} \max_i \alpha^i(s)$, and we have
> \begin{equation}
>     V^{\pi_b}(s_t) \leq \alpha_{\max} \sum^M_{i=1}\sum_a \phi_i(a_t \mid s_t) [r(s_t, a_t) + \gamma E_{s_{t+1} \sim \mathcal{P}(\cdot \mid s_t, a_t)} [V^{\pi_b}(s_{t+1})]] = \alpha_{\max} \text{First Term} +  \alpha_{\max} \text{Second Term}
> \end{equation}
> where,
> \begin{equation}
>  \text{First Term} = \alpha_{\max} E_{a_t \sim \pi^{\zeta_g}(s_t)} [r(s_t, a_t) + \gamma E_{s_{t+1} \sim \mathcal{P}(\cdot \mid s_t, a_t)} [V^{\pi_b}(s_{t+1})]]
> \end{equation}
> \begin{equation}
>  \text{Second Term} = \alpha_{\max}  \sum_{i \neq \zeta_g(s)} \sum_a \phi_i(a_t \mid s_t) [r(s_t, a_t) + \gamma E_{s_{t+1} \sim \mathcal{P}(\cdot \mid s_t, a_t)} [V^{\pi_b}(s_{t+1})]]
> \end{equation}
> Our goal is to extract the term $V^{\pi_{\zeta_g}}(s_t) = E_{{a_{t+i}} \sim \pi^{\zeta_g}(s_{t+i})} [\sum^T_{i=t} \gamma^{i-t} r(s_{t+i}, a_{t+i})]$. It seems to appear in $\text{First Term}$. However, each expansion on $E_{s_{t+1} \sim \mathcal{P}(\cdot \mid s_t, a_t)} [V^{\pi_b}(s_{t+1})]$ leads an additional term, like the second term, inside the expectation of the $\text{First Term}$, making it challenging to induce $V^{\pi_{\zeta_g}}(s_t)$. Consequently, we provide only a non-worse guarantee in Theorem 1.

---

> ### Author Response · Authors · 2024-11-21
> **Reply to W2**
>
> **W2.** We appreciate your suggestion to explore automatic model selection techniques to tune LOM's hyper-parameter.  We have now implemented cross-validation to select the number of Gaussian components $M$ from $\\{1, 5, 10, 15, 20\\}$. This process optimised negative log-likelihood as a measure of goodness of fit.
>
> In the cross-validation process, we employed 3-fold cross-validation, training the GMM model until convergence for $M=\\{1, 5, 10, 15, 20\\}$, and selected the $M$ that minimised the negative log-likelihood on the validation set. We run $3$ different random seeds and report the mean and standard deviation of the model's performance. As shown in the following table, this approach allows us to identify a near-optimal $M$ that enhances model performance while keeping the selection process efficient.
>
> | **Benchmark**              | **M for optimal performance**  | **M for optimal model fit** |
> |----------------------------|----------------------|---------------------------------|
> | Halfcheetah-medium-replay  | $48.8_{\pm 0.7}$     | $46.3_{\pm 0.9}$               |
> | Hopper-medium-replay       | $99.2_{\pm 1.1}$     | $94.2_{\pm 1.0}$               |
> | Walker2d-medium-replay     | $84.8_{\pm 1.0}$     | $82.3_{\pm 1.4}$               |
> | HalfCheetah-full-replay    | $76.6_{\pm 1.2}$     | $76.3_{\pm 2.4}$               |
> | Hopper-full-replay         | $102.0_{\pm 2.7}$    | $100.8_{\pm 1.7}$              |
> | Walker2d-full-replay       | $97.9_{\pm 0.9}$     | $97.4_{\pm 0.5}$               |
>
> It is important to notice that optimal tuning of offline RL parameters in a fully offline manner is still an open problem. All the baselines we have used in our comparison feature one or more hyper-paramaters, which are usually optimised in a similar way, i.e. by on-line tuning. For this reason, we believe that the approach we have taken aligns with current practice and the results we have reported provide a fair comparison. As such, we don't believe that this aspect is a weakness specific to our methodology. Your suggestion has been valuable: as seen in the previous table, optimising a goodness of fit metric produces performance results close to optimal.
>
> We believe that the revised manuscript has become much stronger after incorporating your valuable feedback. We would like to thank you again for raising important questions about LOM. Have we sufficiently addressed the main concerns? Please feel free to let us know if there are additional concerns or questions.

---

> > ### Comment · Reviewer_N1vf · 2024-11-26
> >
> > Thanks for your reply. It has addressed my questions. I will increase my score.

---

> ### Author Response · Authors · 2024-11-25
>
> Hi Reviewer N1vf.
>
> Thank you for your valuable suggestions on our manuscript. We have incorporated your feedback and made improvements to the paper. We kindly ask if you would consider revising the review score. Your further input would also be greatly appreciated.

---

> ### Author Response · Authors · 2024-11-26
>
> Thank you very much for increasing the score! We are glad that we could address your concerns.

---

### Author Response · Authors · 2024-12-02

We would like to sincerely thank all the reviewers and the area chair for their valuable time, insightful comments, and constructive feedback during the review process. Your thoughtful suggestions have greatly contributed to improving the quality of our work, and we deeply appreciate your effort and dedication.

---

### Meta-Review · Area_Chair_J6HX · 2024-12-20

**Metareview:**

This is a strong paper that tackles a relevant problem in offline reinforcement learning. The reviewers found the approach of selectively applying constraints based on mode structure in the behavior data to be novel and effective. The key strength lies in the paper's insightful observation that learning on a single mode can alleviate issues caused by full distribution matching in offline policy improvement. The method of addressing discrete modes and continuous distributions within modes separately is technically sound. While further empirical validation could strengthen the work, the core idea is compelling.  Based on the innovative approach and insightful analysis, I recommend accepting this paper.

**Additional Comments On Reviewer Discussion:**

nothing concerning.

---

### Decision · Program_Chairs · 2025-01-22

Accept (Poster)